# In Situ SEM, TEM, EBSD Characterization of Nucleation and Early Growth of Pure Fe/Pure Al Intermetallic Compounds

**DOI:** 10.3390/ma16176022

**Published:** 2023-09-01

**Authors:** Xiaojun Zhang, Kunyuan Gao, Zhen Wang, Xiuhua Hu, Jianzhu Wang, Zuoren Nie

**Affiliations:** Faculty of Materials and Manufacturing, Beijing University of Technology, Beijing 100124, China; tigerkin1210@163.com (X.Z.); xifengmochou@outlook.com (J.W.); zrnie@bjut.edu.cn (Z.N.)

**Keywords:** pure Fe/pure Al composites, intermetallic compounds, in situ heat treatments, habitus, growth rate

## Abstract

The nucleation and growth processes of pure Fe/pure Al intermetallic compounds (IMCs) during heat treatment at 380 °C and 520 °C were observed through in situ scanning electron microscopy (SEM). The size of the IMCs were statistically analyzed using image analysis software. The types and distribution of IMCs were characterized using transmission electron microscopy (TEM) and electron backscattering diffraction (EBSD). The results showed that: at 380 °C, the primary phase of the Fe/Al composite intermetallic compounds was Fe_4_Al_13_, formed on the Fe side and habituated with Fe. The IMC was completely transformed from the initial Fe_4_Al_13_ to the most stable Fe_2_Al_5_, and the Fe_2_Al_5_ was the habitus with Fe during the process of holding at 380 °C for 15 min to 60 min. At 380 °C, the initial growth rate of the IMC was controlled by reaction, and the growth rate of the thickness and horizontal dimensions was basically the same as 0.02–0.17 μm/min. When the IMC layer thickness reached 4.5 μm, the growth rate of the thickness changed from reaction control to diffusion control and decreased to 0.007 μm/min. After heat treatment at 520 °C (≤20 min), the growth of IMC was still controlled by the reaction, the horizontal growth rate was 0.53 μm/min, the thickness growth rate was 0.23 μm/min, and the main phase of the IMC was the Fe_2_Al_5_ phase at 520 °C/20 min.

## 1. Introduction

Fe/Al composites have the advantages of the excellent mechanical properties of Fe and corrosion resistance, good thermal conductivity, and the low density of Al, so they are more and more widely used in vehicles, ships, power stations, household appliances, and in other fields [1,2,3,4]. However, in the follow-up heat treatment process, the interface of the Fe/Al composites can very easily produce intermetallic phases, which are brittle and can reduce the bonding strength of the interface. Some researchers believe that, in IMCs, a thickness less than 10 μm has no adverse effect on joint strength and may even improve the quality of the joint. However, when the thickness exceeds 10 μm, the joint strength significantly decreases [5,6,7]. Therefore, in the past few decades, in order to control the growth of IMCs at the interface of Fe/Al composites, studies have found that adding elements such as Si and Zn to Al could delay the formation time of IMC at the interface [8,9,10]. Among them, 0.8–1.5% Si element had the best inhibitory effect on IMC formation [11,12,13]; it was used to control Fe/Al intermetallics, inhibit the production of brittle intermetallics [14], and change the distribution and morphology of intermetallic compounds at the interface (such as from continuous distribution to intermittent distribution or from lamellar to spherical distribution) [15]. The interfacial bonding properties of the Fe/Al composites were changed by means of controlling the intermetallic layer thickness below 10 μm [16] and reducing the grain size of the IMC layer [17]. Most of these studies focused on microalloying and IMC growth kinetics. Little attention was paid to the early stages of nucleation and growth of IMCs in Fe/Al systems and their evolution with the increase in the heat treatment temperature or reaction time. However, without a deeper understanding of the entire growth kinetics of these phases, such research would still be an expensive trial-and-error process.

In recent years, in situ experiments have become important tools for elucidating the entire phase transition process at moving interfaces. Szczepaniak [18] used a new type of in situ heating transmission electron microscope for the first time, and the formation process of the Fe_x_Al_y_ phase at the weld interface of a friction stir welding specimen was characterized in real time. The results showed that acicular Fe_4_Al_13_ was the first stable phase formed in the annealed state. Sapanathan [19] characterized the nucleation and early growth stages of Fe/Al intermetallics at 596 °C using an in situ heating device in a special scanning electron microscope with electron backscatter diffraction; the results showed that the Fe_4_Al_13_ phase nucleates first, before the Fe_2_Al_5_ phase diffusion-controlled growth. Barbora Křivská [20] utilized in situ TEM to investigate the formation of Fe_2_Al_5_ at the interface through isothermal annealing above 500 °C. The growth kinetics followed the typical parabolic trend of diffusion-controlled phase transformation. The brittle intermetallic phase that formed reduced the bonding strength between the steel and aluminum. Kai Zhang [21] conducted in situ observations of the melting and solidification process of an Fe/Al/Ta eutectic alloy using high-temperature confocal scanning laser microscopy. The results showed that when the temperature was below 1600 °C, no other types of phase transformations were observed in the Fe/Al/Ta eutectic alloy. After solidification, the strengthening phase exhibited a certain orientation, and the microstructure at the center of the eutectic cell was more regular compared with the microstructure at the grain boundaries. Junsheng Wang [22] conducted a study of the in situ synchrotron radiation imaging of the formation of iron-rich intermetallics during the solidification process of an Al-7.5Si-3.5Cu-0.8Fe (wt.%) alloy. It was found that the nucleation of iron-rich β-intermetallic compounds occurred between 550 and 570 °C. Initially, they grew with an instantaneous tip velocity of 100 μm/s, which then slowed down to 10 μm/s at the end of the growth.

Due to the rapid growth kinetics of IMCs in Fe/Al binary systems, the early stages of nucleation and IMC growth cannot be captured by non-in situ analysis. Therefore, this study employed in situ heating SEM observations at 380 °C and 520 °C to monitor the microstructural and morphological changes in a Fe/Al system, aiming to investigate the nucleation, early growth kinetics, and phase transition of IMC in the initial stages, and quantitative analysis was conducted to study the growth process of the multiple nucleation points of the IMC. The ultimate goal was to explore the growth mechanism of IMC.

## 2. Materials and Methods

The materials were 2 mm annealed pure Al plate (99.99 wt.%) and 3 mm annealed pure Fe plate (99.9 wt.%). Before the rolling composite, the Al plate and Fe plate were pickled, the composite surface was polished with a steel brush, and the rolling deformation was 40%; after the final rolling, the thickness of the Fe/Al composite was 3 mm, the Al layer was 1 mm, and the Fe layer was 2 mm. The cold-rolled composite was carried out at 25 °C; the Fe layer would not oxidize, and the Al layer material would oxidize and produced Al_2_O_3_ after grinding. The thickness of the oxide layer was 2–3 nm, and the oxide layer was broken during the rolling process, which did not adversely affect the rolled composite interface. The wire-cut sample size was 460 × 30 × 1 mm^3^ (length × width × height), and a SEM (Gemini SEM 300 (Zeiss, Oberkochen, Germany)) equipped with a heating table (MINI-HT1200-SE, as shown in Figure 1) was used to heat the sample in situ at 380 °C and 520 °C. The temperature was continuously measured with the thermocouple in contact with the sample, the temperature control accuracy was ±2 °C, and the working voltage of the SEM was 18kV. The microstructure evolution during the heating process was continuously recorded using screen recording software (EVCapture v4.0.2).

In our early non-in situ experiments, it was found that after a 370 °C/1 h heat treatment of pure Fe/pure Al composites, the thickness of IMCs was approximately 10 nm, making it challenging to observe the nucleation and growth process using SEM. These results will be presented in another article. However, after a 380 °C/1 h heat treatment, the IMCs thickness increased to approximately 3–5 μm, which met the size requirements for in situ SEM observation during the heating process. Moreover, at 380 °C, the growth rate of the IMCs layer was slow, which was highly favorable for clear observation of the details of IMCs nucleation and subsequent growth processes. Therefore, 380 °C was selected as the minimum temperature for in situ heating experiments, and 520 °C was the heat treatment temperature of materials used for air cooling in power stations [2].

After the heating stage, because the thickness of intermetallic compound layers fluctuated greatly, in order to better study the growth law of intermetallic compound layers, the maximum values in 20 fields of view were counted in a sample, and the average values were used to characterize the average thickness of intermetallic compound layers, then the intermetallic layer size was evaluated by image analysis system (image pro plus 6.0), FIB sample preparation (Helios G4 PFIB CXe (Thermo Fisher Scientific, Waltham, MA, USA)) was used for TEM (FEITalosF200X-G2 (Lincoln, NE, USA)) microscopic analysis, and after fine polishing and vibration polishing, EBSD analysis (FEI QUANTA FEG650 (Thermo Fisher Scientific, Waltham, MA, USA)) was carried out, as shown in Table 1.

## 3. Results and Discussion

### 3.1. Nucleation and Characterization of Fe/Al Intermetallic Compounds

Figure 2 shows the SEM and TEM characterization of Fe/Al composites intermetallic compounds during in situ heating at 380 °C. Figure 2a shows the interface morphology of Fe/Al composites without heating, Figure 2b shows the enlarged interface morphology of the green box in Figure 2a after heat treatment at 380 °C/20 min. It could be seen from Figure 2b that IMCs of about 100 nm were formed at the interface after heat treatment. The TEM analysis of IMCs was shown in Figure 2c,d. As showed in Figure 2d, the red markings represent the diffraction pattern of Fe, while the green markings represent the diffraction pattern of Fe_4_Al_13,_ and it can be seen that the IMCs were Fe_4_Al_13_ phase, the (2 0 0) plane of Fe was parallel to the (6 0 1) plane of Fe_4_Al_13_, and the axial direction of Fe [0 0 1] was parallel to the axial direction of Fe_4_Al_13_ [1,3,4,5,6], description of the initial Fe_4_Al_13_ and Fe habitus. Therefore, the primary phase of Fe/Al composites intermetallic compounds was the Fe_4_Al_13_.

### 3.2. In Situ Observation and Characterization of Early Growth of IMCs at 380 °C

Figure 3a–i show the nucleation and growth of IMCs during heat treatment at 380 °C, and as can be seen from the picture, IMCs (about 400 nm) began to nucleate at the A position was on the Fe side. With the extension of holding time, three new nucleation points, B, C and D, were formed at the interface, and the thickness was between 100–400 nm. The IMCs at four points had been integrated in the horizontal direction, especially the IMCs at points B and D, which were difficult to distinguish under SEM at 380 °C/60 min. In order to facilitate subsequent size statistics, the horizontal dimension of the IMCs at points B and D was combined into B_width_ + D_width_.

The maximum size of IMCs in the horizontal and thickness direction at points A, B, C and D in Figure 3 were statistically analyzed using image pro, and the growth degree of IMCs at each point was quantitatively measured, as shown in Figure 4. The initial growth rate of IMCs at the four position points was basically consistent in the thickness and horizontal direction. The initial growth rate of IMCs at point A was 0.17 μm/min, point B and point C were all 0.07 μm/min, and point D was 0.02 μm/min. It was worth noting that after 50 min at point A, the growth rate of IMCs in the thickness direction was greatly reduced to 0.007 μm/min, but the growth rate in the horizontal direction had little change.

At the initial stage of IMCs growth, the change in IMCs thickness was controlled by reaction and had a linear relationship with time. In this study, the horizontal size and thickness growth rate were basically the same, which was also linear with time. Therefore, in the initial stage of IMCs growth, the horizontal size of IMCs was also controlled by the reaction. When the thickness of IMCs reached 4.5 μm, the growth rate of thickness decreased greatly, which was due to the limitation of element diffusion, and the growth mode changed from reaction control to diffusion control, so, at 380 °C, the critical thickness of the transition from the reaction-controlled to diffusion-controlled growth rate was 4.5 μm.

When the second phase particles were very small, they had a high surface-to-volume ratio and a small curvature radius. In this case, the influence of surface tension on solubility must be considered, which is known as the Gibbs–Thomson effect [23]. Indeed, the corrected solubility limit *C_r_* of B atoms in α matrix in equilibrium with β phase occurring as spherical particles of radius *r* is often given as a function of *r* [24]:(1)$$C_{r}^{\alpha }{=C}_{\infty }^{\alpha }exp(\frac{{2\gamma V}_{m}}{rRT})$$
where *C_r_* is the solute concentration at the interfacial region in the matrix near the second phase. *r* is the radius of the second phase particles. *T* is the temperature, *γ* is the surface energy, *R* is the molar gas constant and *V_m_* is the molar volume.

Considering the Gibbs–Thomson effect, it was demonstrated in Figure 4b schematically the solute concentration *C_r_* with changing of particle radius. Because the radii of the second phase particles at four positions pointed in Figure 3 was in the order of *r_A_* > *r_B_* (or *r_C_*) > *r_D_*, the solute concentration near the particles could be estimated based on Equation (1) as *C_rD_* > *C_rB_* (or *C_rC_*) > *C_rA_*. It is clear that the solute concentration in the Al matrix was 100%, thereafter the solute concentration difference between *C_r_* and Al matrix was Δ*C_r_A* > Δ*C_rB_* (or Δ*C_rC_*) > Δ*C_rD_*_._ Correspondingly, the chemical driving force for the second phase was greatest at point A and least at point D; therefore, the growth rates of the second phase at the four points were evaluated to be *k_A_* > *k_B_* (or *k_C_*) > *k_D_*.

It should be noted that the surface of the sample heated in situ observed in this study was an open free surface, which was easier to nucleate and grow than the inside of the sample, and its morphology was irregular polygon-like. In the subsequent TEM and EBSD characterization, it was found that the size of IMCs on the non-free surface inside the interface was significantly smaller than that on the free surface, and its morphology was flat and its thickness was evenly distributed.

Figure 5 shows the TEM analysis of the interface IMCs after heat treatment at 380 °C/60 min, and Figure 5a shows the interface morphology of the sample after fine grinding and polishing. The thickness of the IMCs layer was about 500 nm, and the orange box represented the FIB sample preparation area. Figure 5b shows the sample prepared by FIB, and it can be seen that there are two obvious pore defects in the IMCs layer. IMCs was closely bound to the contact surface of Fe layer and Al layer, without obvious gaps, cracks, and other defects. Figure 5c shows the TEM bright field diagram, and it can be seen that cracks and other defects appeared at the contact surface between the IMCs and Fe layer, as well as between the IMCs and Al layer, which should be generated during the thinning process of the FIB sample preparation and not the defects of the sample itself. The diffraction analysis of IMCs was shown in Figure 5f, the IMCs were polycrystalline Fe_2_Al_5_. We performed high-resolution analysis on the interface between IMCs layer and Fe layer (green color box in Figure 5c) and the interface between IMCs layer and Al layer (red color box in Figure 5c), respectively, and the results showed that the (2 1 1) and (2 2 0) planes of the Fe_2_Al_5_ phase were parallel to the (1 0 $\bar{1}$) and (1 1 0) planes of Fe, respectively, and Fe_2_Al_5_ was habituated with Fe, but no parallel plane existed between Fe_2_Al_5_ and Al.

According to reports [5,6,7], when the thickness of IMCs was less than 4 μm, and no cracks or other defects on the contact surface of the IMCs and Fe layer, IMCs and Al layer, the bonding strength of IMCs relative to Fe/Al composites interface had no adverse effects, in this study, at 380 °C/60 min, the IMCs were about 500 nm, and there were no cracks and other defects on the contact surface, so the IMCs had no adverse effect on the bonding strength of the Fe/Al composites interface. The IMCs of Fe/Al composites were completely transformed from the initial Fe_4_Al_13_ to the most stable Fe_2_Al_5_ during heat treatment at 380 °C for 15 min to 60 min. Both the Fe_4_Al_13_ and Fe_2_Al_5_ phases were habituated with Fe, indicating that Fe_4_Al_13_ and Fe_2_Al_5_ phases had a closer orientation to Fe and were generated from the Fe side.

### 3.3. In Situ Observation and Characterization of IMCs at 520 °C

Figure 6a–i showed the nucleation and growth of IMCs during heat treatment at 520 °C; as seen from Figure 6b, when the holding time was 4 min, the IMCs began to nucleate at the interface, and the IMCs were all less than 1 μm, at the same time, the horizontal size of IMCs on the left side of point A was L_width_. When the holding time was extended to 6 min, more nucleation occurred at the interface, and obvious crack defects were formed on the Al side near the IMCs layer, as indicated by the yellow arrow in Figure 6c–i. With the extension of holding time, the size of IMCs gradually increased along the thickness and horizontal direction, and the cracks and defects concentrated in the Al side near the IMCs increased. The average thickness size and L_width_ of the interface IMCs in the whole process were statistically analyzed, and the results were shown in Figure 7.

It could be seen from Figure 7 that the average thickness growth rate of IMCs was 0.23 μm/min, and the growth rate of L_width_ was 0.53 μm/min, which was 2.3 times larger than that of the thickness. Compared with 380 °C, at 520 °C heat treatment, the growth of IMCs phase size was still controlled by the reaction, but its growth rate in the horizontal direction was significantly higher than that in the thickness direction.

There were obvious crack defects on the Al side, which might be caused by two reasons [25]: ① The Kirkendall effect during the diffusion of elements; ② The volume expansion caused by IMCs generation. A single Al cell was composed of 4 Al atoms, the volume of the cell was V_Al_ = 66.3 Å^3^, a single Fe cell consisted of 2 Fe atoms, and the cell volume V_Fe_ = 23.6 Å^3^, a single Fe_2_Al_5_ cell consisted of 10.8 Al atoms and 4 Fe atoms, and the cell volume V_Fe2Al5_ = 206.5 Å^3^. Because the difference between V_Fe2Al5_ and the consumed V was less than 10%, the volume change in the generated Fe_2_Al_5_ phase was not the main reason for the crack defects.

At 520 °C, D_Al_ (Fe_2_Al_5_) = 8.1 × 10^−6^ cm^2^/s > D_Fe_ (Fe_2_Al_5_) = 3.6 × 10^−12^ cm^2^/s [26], the number of atoms of Al atom arriving at the Fe interface through Fe_2_Al_5_ was six orders of magnitude higher than that of Fe atom arriving at the Al interface through Fe_2_Al_5_ phase, so the number of diffused Fe atoms was negligible, and the diffusion of Al in Fe_2_Al_5_ was dominant. In the subsequent process of this experiment (at 520 °C/60 min), the crack defects would be transformed into separated cracks, as the sample became more continuous during cooling (the results of this part of the experiment will be shown in another article), and it could be inferred that the crack defects were mainly caused by the Al side of Kirkendall.

Figure 8 shows the EBSD characterization of Fe/Al composites after in situ heating at 520 °C/20 min, RD surface with fine grinding and vibration polishing. It can be seen that the main constituent phase of IMCs was Fe_2_Al_5_, and only a small amount of Fe_4_Al_13_ was distributed in the middle of Fe_2_Al_5_. It was confirmed that Fe_2_Al_5_ was the main phase and stable phase of IMCs in Fe/Al composites.

During the heat treatment process, the Fe_4_Al_13_ was the primary phase that formed at the Fe/Al interface initially; subsequently, the Fe_4_Al_13_ underwent rapid reaction with the Fe, transforming into the Fe_2_Al_5_; therefore, the Fe_2_Al_5_ rapidly grew and became the dominant phase of IMCs. Moreover, the diffusion rate of Al within the Fe_2_Al_5_ was significantly higher than that of Fe within the Fe_2_Al_5_ [26], the new Fe_4_Al_13_ forms at the Al/Fe_2_Al_5_ interface [27], the coexistence of two IMCs, with the Fe_2_Al_5_ as the dominant one, had been observed. In Figure 8b, an inverse pole figure (IPF) map was presented, showing that the Al matrix was in a recrystallized state, while the Fe matrix was still in a deformed state. Concerning Fe_2_Al_5_, at 520 °C, the reaction might locate at the proper value region of driving force and diffusion rate, i.e., the nose of the reaction, which could result in smaller grain sizes.

## 4. Conclusions

(1)At 380 °C, the primary phase of Fe/Al composite intermetallic compounds was Fe_4_Al_13_ formed on the Fe side and habituated with Fe. The IMCs changed from the initial Fe_4_Al_13_ to the most stable Fe_2_Al_5_ when the heat treatment was extended from 15 min to 60 min, and the Fe_2_Al_5_ phase was habitus with Fe.(2)During heat treatment at 380 °C, the initial growth rate of Fe/Al composite intermetallic compounds was controlled by reaction. The initial growth rate of thickness and horizontal dimensions of IMCs was basically the same, ranging from 0.02–0.17 μm/min. When the thickness reached 4.5 μm, the growth rate of the thickness changed from reaction control to diffusion control, the critical thickness of the transformation was 4.5 μm, and the growth rate decreased to 0.007 μm/min.(3)After heat treatment at 520 °C (≤20 min), the growth of IMCs was still controlled by the reaction, the horizontal growth rate was 0.53 μm/min, the thickness growth rate was 0.23 μm/min, and the main component of IMCs was Fe_2_Al_5_ at 520 °C/20 min.

## Figures and Tables

**Figure 1 materials-16-06022-f001:**
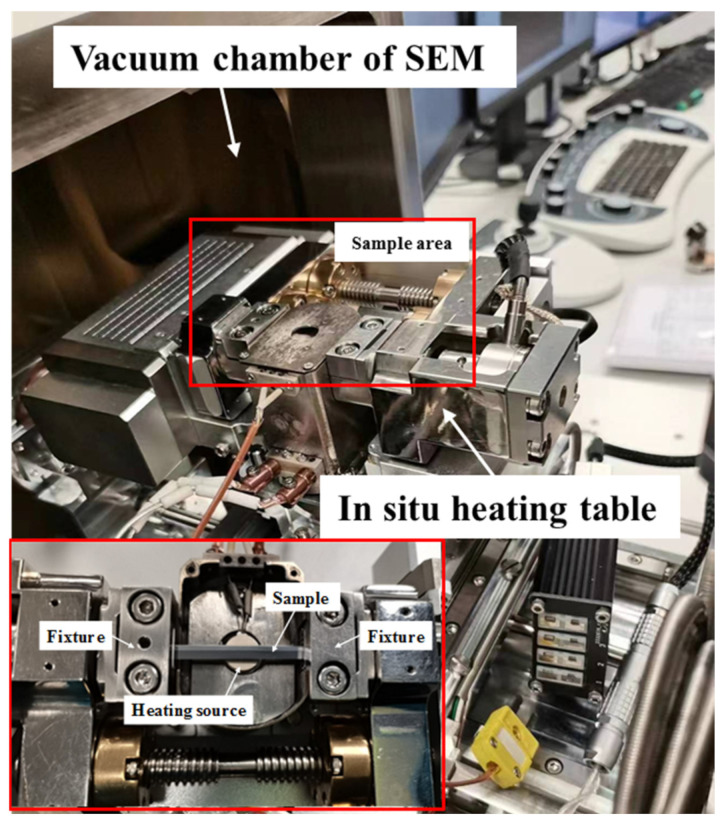
In situ heating unit.

**Figure 2 materials-16-06022-f002:**
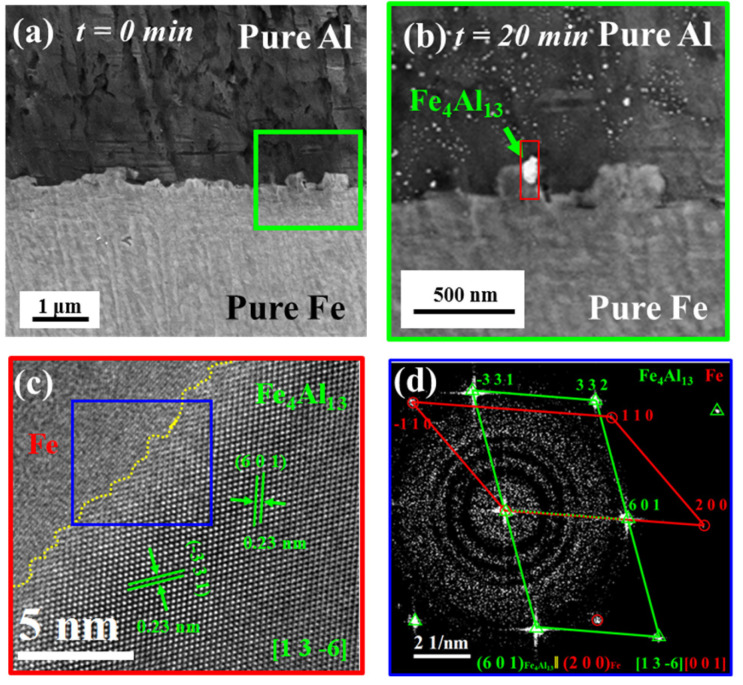
Fe/Al in situ heat treatment with SEM and TEM at 380 °C/20 min. (**a**) In situ SEM of Fe/Al composites at 380 °C/0 min. (**b**) In situ SEM of Fe/Al composites at 380 °C/20 min. (**c**,**d**) TEM of Fe/Al composites at 380 °C/20 min.

**Figure 3 materials-16-06022-f003:**
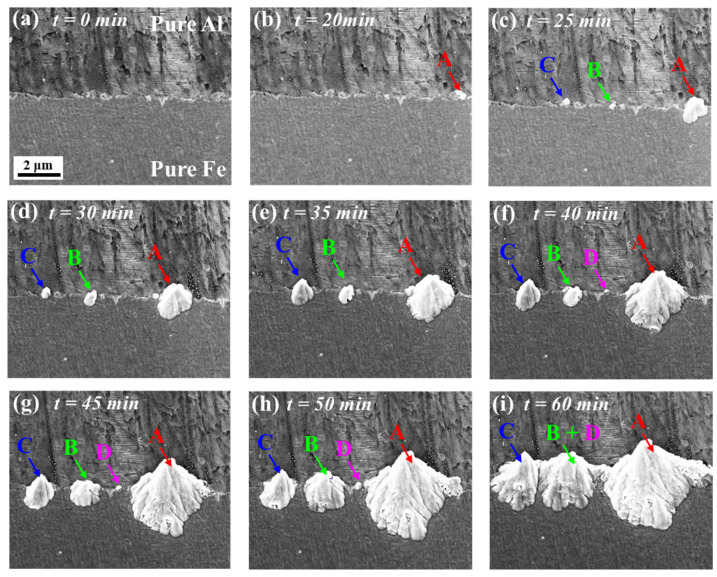
Sequence of IMCs nucleation and growth obtained from in situ SEM observations at 380 °C (**a**) *t* = 0 min; (**b**) *t* = 20 min, the red arrow indicates where the interface IMCs began to nucleate and marked it as A; (**c**) *t* = 25 min, two nucleation points B and C were added at the interface; (**d**) *t* = 30 min; (**e**) *t* = 35 min; (**f**) *t* = 40 min, a D-nucleate point was added at the interface; (**g**) *t* = 45 min; (**h**) *t* = 50 min; (**i**) *t* = 60 min, the IMCs of point B and D grew into one body, and then the IMCs of two points B and D were analyzed as B + D.

**Figure 4 materials-16-06022-f004:**
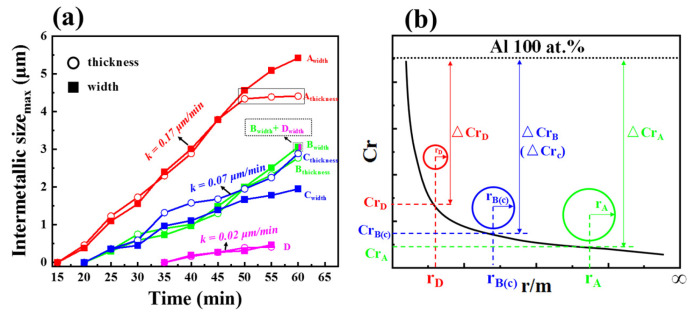
(**a**) Evolution of the maximum size of IMCs phase in horizontal direction and thickness direction over time at points A, B, C and D (in Figure 3); (**b**) the Cr with changing of r (considering the Gibbs–Thomson effect).

**Figure 5 materials-16-06022-f005:**
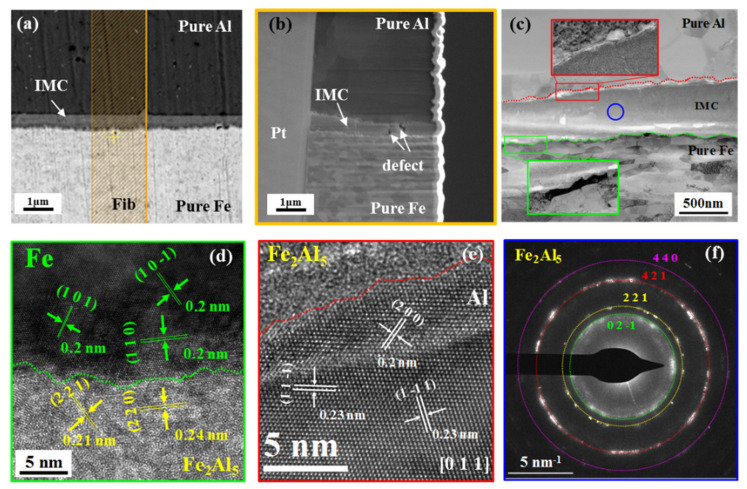
TEM analysis of IMCs after heat treatment at 380 °C/60 min. (**a**) SEM interface morphology of IMCs after fine grinding and polishing; (**b**) SEM morphology after FIB preparation; (**c**) TEM sample open field diagram; (**d**) high resolution image at the interface between IMCs layer and Fe layer; (**e**) high resolution image at the interface between IMCs layer and Al layer; (**f**) diffraction pattern of IMCs phase.

**Figure 6 materials-16-06022-f006:**
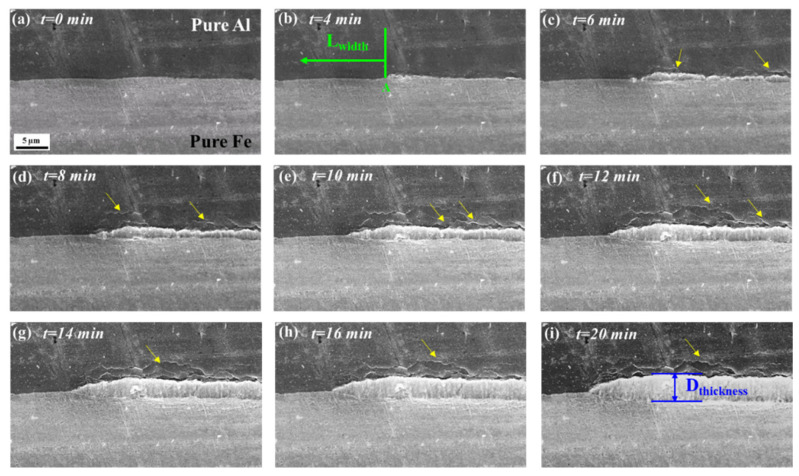
Sequence of IMCs nucleation and growth obtained from in situ SEM observations at 520 °C (**a**) *t* = 0 min; (**b**) *t* = 4 min, the horizontal dimension of the IMCs layer to the left of point A was L_width_; (**c**) *t* = 6 min; (**d**) *t* = 8 min; (**e**) *t* = 10 min; (**f**) *t* = 12 min; (**g**) *t* = 14 min; (**h**) *t* = 16 min; (**i**) *t* = 20 min.

**Figure 7 materials-16-06022-f007:**
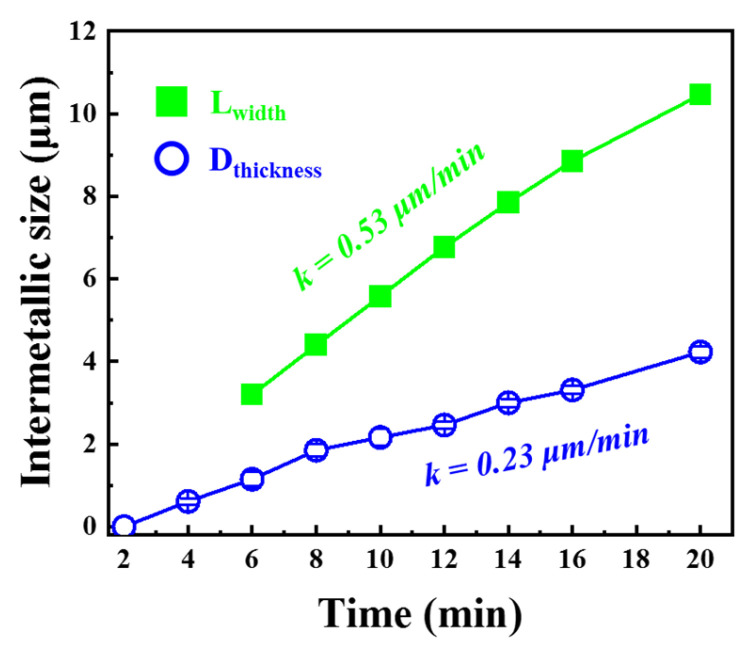
Evolution of IMCs phase average thickness and maximum L_width_ size over time.

**Figure 8 materials-16-06022-f008:**
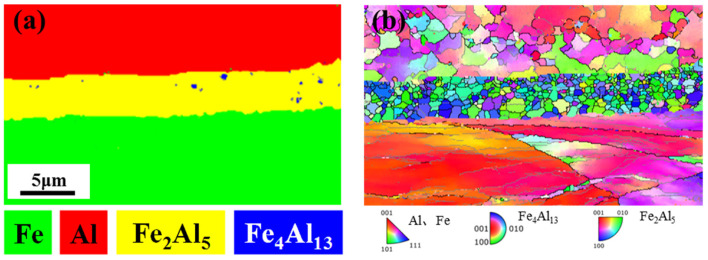
(**a**,**b**) EBSD characterization of IMCs at 520 °C/20 min.

**Table 1 materials-16-06022-t001:** Samples in materials and methods.

Sample	Materials	Temperature/°C	Holding Time/min	Characterization Methods
1#	pure Fe/pure Al	380	20	FIB + TEM
2#		380	60	FIB + TEM, Size statistics
3#		520	20	EBSD, Size statistics

## Data Availability

This study did not report any data.

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
