# Peer review of "In Situ SEM, TEM, EBSD Characterization of Nucleation and Early Growth of Pure Fe/Pure Al Intermetallic Compounds"

_materials, 2023, doi:10.3390/ma16176022_

Round 1
Reviewer 1 Report
I have now gone through the manuscript carefully. Despite interesting results on bi-metallic Al-Fe materials, I do not find that this manuscript proposes sufficient advances to this field, or brings a really new perspective on the understanding of relationships between processing, microstructure, and properties that a reader can expect. Furthermore, the discussion of data is really poor.Author Response
Please see the attachment.

Reviewer 2 Report
This works studies the nucleation and early growth of AlFe intermetallics with the employment of in-situ SEM, TEM and EBSD. This work is novel and should be considered for publication. Please find a few comments below:
1) Fe-Al intermetallics have been studied in detail. What is the novelty of this work? You are kindly requested to expand the discussion.
2) Authors repeat "pure Fe/pure Al intermetallics". Please remove pure.
3) Authors explained why they chose 520 C but didn not explain in detail why they chose 380 C for the growth study.
4) In figure 4 there are big differences in the growth rates, this needs to be explained and expand the discussion on growth rates.
5) There is a big difference in grain sizes in figure 8. What is the reason? Expand the discussion. What is the reason for the existance of 2 different types of AlFe intermetallics?
Reviewer 3 Report
It is an interesting paper related to Fe/Al inter metallic compounds.
What happens with references 1 to 4?, they are not cited in the text.
Figure 2: revise the units of 0.23....they are micras or nm?
Lines 144, 146, 156: Revise "Fe Al", it seems that there is some typing mistake
I don´t understand what's meaning "[J]" in some references
Reference 18 is not complete

Reviewer 4 Report
1) Abstract can be strengthened by incorporating the material and method used for the research work.
2) The novelty of the work is to be mentioned at the end of Introduction section.
3) How the composite material dimension and thickness is arrived? Is it in accordance with the convenience, standard or literature review?
4) While rolling of the composite, what care is taken on pure aluminium to prevent possible oxidation into Al2O3? Please include in the revised manuscript
5) In some of the Reference title of the paper, the first letter of each word is of upper-case letter whereas in some cases only first word’s first letter is of upper-case letter. For format uniformity, please modify.
Minor revision is required
Round 2
Reviewer 1 Report
The efforts of the Authors to improve the manuscript have been really appreciated. The paper is now suitable to be published in Materials.